# A Comparative Study of Serum Angiogenic Biomarkers in Cirrhosis and Hepatocellular Carcinoma

**DOI:** 10.3390/cancers14010011

**Published:** 2021-12-21

**Authors:** Krizia Pocino, Cecilia Napodano, Mariapaola Marino, Riccardo Di Santo, Luca Miele, Nicoletta De Matthaeis, Francesca Gulli, Raffaele Saporito, Gian Ludovico Rapaccini, Gabriele Ciasca, Umberto Basile

**Affiliations:** 1Unità Operativa Complessa di Patologia Clinica, Ospedale Generale di Zona San Pietro Fatebenefratelli, 00189 Rome, Italy; krizia.pocino@gmail.com (K.P.); saporito.raffaele@fbfrm.it (R.S.); 2Synlab Data Medica, 35133 Padova, Italy; cecilia.napodano@gmail.com; 3Dipartimento di Medicina e Chirurgia Traslazionale, Sezione di Patologia Generale, Università Cattolica del Sacro Cuore, Fondazione Policlinico Universitario “A. Gemelli” IRCCS, 00168 Rome, Italy; 4Dipartimento di Neuroscienze, Sezione di Fisica, Università Cattolica del Sacro Cuore, Fondazione Policlinico Universitario “A. Gemelli” IRCCS, 00168 Rome, Italy; riccardo.disanto92@gmail.com (R.D.S.); gabriele.ciasca@unicatt.it (G.C.); 5Dipartimento di Medicina e Chirurgia Traslazionale, Sezione di Medicina Interna, Università Cattolica del Sacro Cuore, Fondazione Policlinico Universitario “A. Gemelli” IRCCS, 00168 Rome, Italy; luca.miele@unicatt.it (L.M.); nicoletta.dematthaeis@policlinicogemelli.it (N.D.M.); gianludovico.rapaccini@policlinicogemelli.it (G.L.R.); 6Laboratorio di Patologia Clinica, Ospedale Madre Giuseppina Vannini, 00177 Rome, Italy; dottfgulli@gmail.com; 7Dipartimento di Scienze di Laboratorio e Infettivologiche, Fondazione Policlinico Universitario “A. Gemelli” IRCCS, 00168 Rome, Italy

**Keywords:** hepatocellular carcinoma, cirrhosis, neoangiogenesis factors

## Abstract

**Simple Summary:**

The progression of liver disease is accompanied by pathological angiogenesis, a prerequisite for the development of HCC. In this paper, we analyzed the clinical significance of serum angiogenic markers VEGF, Ang-1, Ang-2, angiopoietin receptor Tie1/2, HGF, and PECAM-1 in 62 patients with liver disease, out of which 33 were diagnosed with HCC and 29 with liver cirrhosis without signs of neoplasia. Biomarkers levels were investigated as a function of “Model for End-Stage Liver Disease” (MELD) score and Fibrosis Index (FI). HCC patients showed higher HGF levels than ones with cirrhosis, while high Ang-1 levels appeared to have a protective role in HCC as well as prognostic significance; we also found a strong correlation between HGF levels, Ang-2, and VEGF levels, further supporting their role in tumor angiogenesis. Due to the complexity of angiogenesis and the small size of the study group, further investigations are widely desired especially in the era of immunotherapy and HCC-targeted anti-angiogenic drugs.

**Abstract:**

Background: Hepatocellular carcinoma (HCC) is a global health problem associated with chronic liver disease. Its pathogenesis varies according to the underlying etiological factors, although in most cases it develops from liver cirrhosis. The disease progression is accompanied by pathological angiogenesis, which is a prerequisite that favors the development of HCC. Aims: This study aims at contributing to our understanding of the role of angiogenic factors in the progression of liver disease. For this purpose, we evaluate the clinical significance of serum angiogenic markers (VEGF, Ang-1, Ang-2, the angiopoietin receptor Tie1/2, HGF, and PECAM-1) first in cirrhotic and HCC patients separately, and then comparing cirrhotic patients with and without HCC. Materials and Methods: We enrolled 62 patients, out of whom 33 were diagnosed with HCC and 29 with liver cirrhosis without signs of neoplasia. Patients underwent venous blood sampling before and after receiving treatments for the diagnosed disease. Serum markers were evaluated using ELISA assays for Tie1 and the Bio-Plex Multiplex system for the remaining ones. Biomarker levels were investigated as a function of clinical scores for disease staging (MELD and Fibrosis Index, FI). Results: In cirrhotic patients, Ang-1 and Ang-2 correlate with MELD (ρ_Ang-1_ = −0.73, *p* = 2E−5) and FI (ρ_Ang-1_ = −0.52, *p* = 7E−3, ρ_Ang-2_ = 0.53, *p* = 3E−3). A reduction of Ang-2 levels (*p* = 0.047) and of the Ang-2/Ang-1 ratio (*p* = 0.031) is observed in cirrhotic patients diagnosed with viral hepatitis after antiviral treatments. In HCC patients, Ang-1 negatively correlates with FI (ρ = −0.63, *p* = 1E−4), and PECAM-1 positively correlates with MELD (ρ = 0.44, *p* = 0.01). A significant Ang-1 reduction was observed in deceased patients during the study compared to ones who survived (*p* = 0.01). In HCC patients, VEGF levels were increased after tumor treatment (*p* = 0.037). Notably, HGF levels in cirrhotic patients with HCC are significantly raised (*p* = 0.017) compared to that in those without HCC. Conclusions: Our results suggest that serum angiogenic markers, with emphasis on Ang-1/2, can contribute to the development of quantitative tools for liver disease staging and therapy monitoring. The comparison between cirrhotic patients with and without HCC suggests that HGF levels are potentially useful for monitoring the insurgence of HCC after a cirrhosis diagnosis. High Ang-1 levels in HCC patients appear to have a protective role as well as prognostic significance.

## 1. Introduction

Hepatocellular carcinoma (HCC) is a global health problem associated with chronic liver disease; it represents the second leading cause of cancer death in males and the sixth in females (males/females ~2:1), with an annual incidence of 13,000 new cases (in 2020) in Italy, 3% of all new cases of cancer [1]. The pre-existence of risk factors is associated with over 70% of primary liver tumors, mainly related to the prevalence of hepatitis C virus (HCV) infection. Hepatitis B virus (HBV) infection is also related to the onset of the disease, with a prevalence in Asia and Africa, while in the other countries its role is predictably destined to decrease because of vaccination campaigns from 1978 onward [2]. Among non-infective risk factors, the aflatoxins ingested through food and alcohol consumption play an important role [3,4].

The pathogenesis of chronic liver disease varies according to the underlying etiological factor, although in most cases it develops from liver cirrhosis. Crucial steps toward worsening progression and carcinogenesis include chronic inflammation, alterations in the tumor micro- and macroenvironment, and angiogenesis. Both intrinsic individual genetic predisposition and extrinsic risk factors can lead to the development of HCC [5].

The worsening progression of liver disease is accompanied by pathological angiogenesis, which is a prerequisite that favors the development of HCC. Angiogenesis takes place through different progressive steps and represents the limiting factor for the speed of tumor growth [6]. The growth of avascular tumors is limited by the distance from near vessels for the uptake of oxygen, nutrients, and the discharge of catabolic products through the interstitium. Therefore, an angiogenic “switch”, through the production of angiogenic factors, is a necessary feature of a tumor that can grow [7]. In normal situations, there is a balance between endogenous angiogenic inducers and endogenous angiogenic inhibitors that keeps the angiogenic process under control and prevents inappropriate tissue vascularization. Angiogenesis inhibitors are often derived from circulating extracellular matrix proteins (as a result of injury to the matrix), e.g., fibronectin, prolactin, collagen XVIII (endostatin), NK1 fragment of the hepatocyte growth factor (HGF), and angiostatin [8]. Virtually all endogenous angiogenesis inhibitors suppress tumor growth in animal models.

Vascular endothelial growth factor (VEGF) is best known as the most potent stimulator of normal and pathological angiogenesis. VEGF release increases under hypoxic conditions [9,10]. Its expression is regulated by the inducible factor of hypoxia (HIF-1a), which triggers the VEGF transcription [11]. This indicates that VEGF participates in the initial phase of angiogenesis. As a matter of fact, the transition of endothelial cells from an inactive to an active state can occur along with their proliferation, migration, and formation of new vessels.

The tyrosine kinase receptors (Tie1 and Tie2) and their angiopoietin 1–4 ligands (Ang1, -2, -3, and -4) play a key role during the late phase of angiogenesis and are responsible for the maturation of newly established vascular structures. Ang1 and Ang2 are the best described and characterized angiopoietins [12]. The activity of the angiopoietin/Tie system determines the stabilization of new vessels. Both Ang1 and Ang2 interact with the same Tie2 receptor site having a similar affinity toward it, but only Ang1 induces its phosphorylation and the subsequent activation [13].

There is growing evidence that the angiopoietin/Tie signal can influence the outcome of inflammation [14]. Ang1 appears to be a powerful activator of Tie2, as well as a regulator of blood vessel formation and maturation. Experimental studies have shown that Ang1 acts as an anti-inflammatory molecule [15], but it can induce significant complications such as pulmonary hypertension [16]. Ang1 neutralizes tissue factor (TF) activity that is relevant for the negative control of coagulation, thrombosis, and inflammatory response. Furthermore, Ang1 reduces the adhesion of VEGF-related leukocytes to the endothelium [17,18]. On the contrary, Ang2 acts as a competitive antagonist of Ang1, deregulates the signal pathway of Tie2 [13], and exerts pro-inflammatory effects [19,20]. Additionally, significantly elevated serum Ang2 levels have been observed during carcinogenesis in HCC patients [21].

It has been shown that HGF is overexpressed in HCC compared to the normal liver [22,23]. Stellate cells and myofibroblasts are induced to secrete HGF from tumor cell products, and HGF, in turn, stimulates the invasiveness of tumor cells [24]. Recently published data show that higher HGF serum levels negatively correlate with patient survival time [25] and positively with tumor size [26,27].

PECAM-1 (platelet endothelial cell adhesion molecule-1) also known as CD31 is normally expressed on the surface of endothelial cells, platelets, leukocyte subpopulations, and Kupffer cells [28]: these intercellular interactions are crucial for the angiogenesis process. In this context, PECAM-1 mediates both homophilic and heterophilic adhesion [29]. Its identification can help in assessing the degree of tumor angiogenesis, which may indicate a rapidly growing tumor [30].

Starting from this knowledge, the mentioned angiogenic markers appear to have a great potential for the development of liquid-biopsy approaches useful for monitoring subjects at high risk of HCC. Toward this goal, here, we evaluate the possible clinical utility of circulating angiogenic markers (VEGF, Ang-1, Ang-2, the angiopoietin receptor (Tie1/2), HGF, and PECAM-1), first in cirrhotic and HCC patients separately, and then comparing cirrhotic patients with and without HCC. In both cohorts, serum levels of these biomarkers are studied as a function of widely utilized clinical scores to verify their applicability for disease staging, and before and after patients’ treatment to test their effectiveness for therapy monitoring. Finally, the comparison between the measured concentrations of angiogenic markers in cirrhotic patients with and without HCC is meant to assess whether these parameters have possible applications in keeping cirrhotic subjects who are at high risk of developing HCC under close surveillance.

## 2. Material and Methods

### 2.1. Patients

This is a non-profit interventional study, that involved the recruitment of patients belonging to the Liver Diseases Outpatient Clinic, suffering from liver cirrhosis of different etiology and/or HCC. The subjects under examination underwent peripheral blood sampling to measure circulating levels of the main neoangiogenesis factors as biomarkers of carcinogenesis.

For all the recruited cirrhotic and HCC subjects, the first blood draw (an 8 mL tube of serum) was sampled on the day of hospital admission. Then, all subjects received a diagnosis and undertook normal outpatient clinical monitoring, which varied according to the type, stage, and treatment of the disease. The second blood sample for cirrhotic patients diagnosed with viral hepatitis (HCV/HBV), either treated with direct-acting antiviral agents (DAAs) or HBV conventional antiviral drugs, was acquired between 8 and 24 weeks after sustained virologic response (SVR), depending on the genotype and the ongoing patient response. For cirrhotic patients with different underlying diseases, the second blood sample was not evaluated in the present study. Differently, for HCC patients, neoangiogenesis factors were re-evaluated at scheduled controls after treatments, except for patients receiving the best supportive care (BSC), for whom these factors were not evaluated in this phase of the experiment. Controls were done at different times, according to international guidelines [31], for the different treatments, both in the presence and absence of recurrences. Treatments included percutaneous alcoholization (PEI), radiofrequency ablation (RFA), intra-arterial chemoembolization (TACE), intra-arterial radioembolization (TARE), surgical resection, transplantation, and systemic antiangiogenic treatment. The second blood sample was taken for patients treated with locoregional therapies (PEI, RFA, TACE, and TARE) at the first imaging procedure (CT with contrast medium) documenting the absence of neoplastic tissue in the tumor; for patients treated by surgery (resection or transplantation), at the fourth month from the surgical procedure; and for patients treated with antiangiogenic drugs, at the first imaging examination (CT with contrast medium) documenting a reduction of neoplastic tissue in the target lesion. See the schematic workflow in Appendix A.

The inclusion criteria were as follows: patients aged 18 years or older; patients with liver cirrhosis of different etiology; patients with liver cancer at diagnosis; patients who gave written informed consent.

The exclusion criteria were as follows: the presence of infections other than HCV and HBV; severe comorbidities at the time of enrollment; participation in other clinical trials involving the use of drugs; pregnant women.

Among the 82 subjects evaluated for this study, we enrolled 62 patients (35 men and 27 women aged between 26 and 85 years) who met all the inclusion criteria: 33 out of 62 subjects were diagnosed with HCC and 29/62 with liver cirrhosis of different etiology without signs of neoplasia. Severity of the liver disease was assessed, for each patient, by estimating two widely used clinical scores, i.e., the “Model for End-Stage Liver Disease” MELD score (MELD = 3.78 × ln[serum bilirubin (mg/dL)] + 11.2 × ln[INR] + 9.57 × ln[serum creatinine (mg/dL)] + 6.43) and the Fibrosis Index (FI = 8.0 − 0.01 × Plt (10^3^/μL) - Alb (g/dL)) [32,33]. Patient demographic and clinical characteristics, including comorbidities, are summarized in Table 1. Regarding comorbidities, portal hypertension was evaluated with the portal venous pressure gradient (HVPG) measurement [34]. Values higher than 10 mmHg were considered indicative of portal hypertension with the indication to perform endoscopy of the upper gastrointestinal tract for the evaluation of the presence and degree of esophageal varices. The esophageal varices were classified based on the Paquet classification (Grade 1–Grade 3) [35]. Patients with diabetes were all diagnosed with T2DM, compensated, not insulin-dependent, and treated with oral agents.

### 2.2. Laboratory Procedures

For the measurement of Ang-1, Ang-2, VEGF, Tie1/2, HGF, and PECAM-1 serum levels, serum samples were centrifuged at 2000 g/min for 15 min and stored at −80 °C until their use. The assessment of these biomarkers was carried out at the Laboratory of Clinical Immunology and Molecular Hepatology of the Department of Medical Sciences of the Fondazione Policlinico Universitario “A. Gemelli”—I.R.C.C.S (Rome), and at the Institute of General Pathology of the Università Cattolica del Sacro Cuore (Rome), using ELISA assays (FineTest^®^, Wuhan Fine Biotech Co., Ltd., Wuhan, Hubei, China) for Tie1 and the Bio-Plex Multiplex (BIO-RAD, Hercules, CA, USA) system for Ang-1, Ang-2, VEGF, Tie2, HGF, and PECAM-1.

AST, ALT, GGT, CHOL, HDL, LDL, TG, ALP, AFP, and CA 19-9 were measured during the clinical routine by ADVIA Centaur instruments (Siemens Healthcare GmbH, Munich, Germany).

All the blood tests (angiogenic and conventional biochemical parameters) were performed in a single analytical session, following the instructions provided by the manufacturers, and the determinations were performed by an operator without knowledge of the clinical information of the handled sample. Each sample was tested twice to minimize eventual discrepancies, and all tests were performed in the same laboratory with the same instruments.

The following abbreviations were used for angiogenic markers. HGF: hepatocyte growth factor; VEGF: vascular endothelial growth factor; PECAM-1: platelet endothelial cell adhesion molecule-1; Tie: tyrosine kinase receptors.

The following abbreviations were used for the remaining laboratory parameters. AST: aspartate aminotransferase; ALT: alanine aminotransferase; GGT: gamma-glutamyl transferase; CHOL: cholesterol; HDL: high-density lipoprotein; LDL: low-density lipoprotein; TG: triglycerides; ALP: alkaline phosphatase; AFP: alpha-fetoprotein; CA 19-9: carbohydrate antigen 19-9; WBC: white blood cells.

### 2.3. Ethical Consideration

The ethic committee of our institution (Fondazione Policlinico Universitario “A. Gemelli” I.R.C.C.S., Università Cattolica del Sacro Cuore) approved the study (Protocol ID: 2078). All patients gave written informed consent to the use of their clinical and serological data in this study. The whole study was conducted according to the Declaration of Helsinki, as revised in 2013.

### 2.4. Statistical Analysis

The continuous variables were tested to verify their normality by analyzing the QQ plot and the Shapiro–Wilk test (data not shown). Some of the analyzed biomarkers showed significant deviations from normal. Therefore, unless explicitly stated, the database was analyzed using non-parametric methods and comparisons between groups were performed with the Wilcoxon unpaired two-sample test. First, the presence of correlations among the various plasma biomarkers considered and some basic characteristics of the patients, such as age, MELD score, and FI, was evaluated. For this purpose, the Spearman correlation index (ρ) was estimated for each pair of variables of interest. The ρ values were calculated using the programming language R, with reference to the algorithms implemented in the corrPlot function of the Hmisch package. The statistical significance of the correlation coefficients was evaluated using a power analysis. The obtained values were summarized in a correlation matrix, according to Wei and coworkers [36]. A color code and the size of the points of the map were jointly used to allow immediate assessment of the strength of the correlation and its direction. The relationship between selected continuous variables was also investigated through univariate linear regression.

Finally, the performance of the investigated markers in distinguishing the two groups (HCC patients deceased and not deceased) was assessed by logistic regression followed by ROC curve analysis. ROC curves and AUC values were calculated as described [37,38] and using the R package pROC [39].

## 3. Results

### 3.1. Analysis of Angiogenic Biomarkers in Cirrhotic Patients and Correlation with Clinical Parameters

In this section, we describe the investigation of the clinical significance of selected serum angiogenesis biomarkers (VEGF, Ang-1, Ang-2, the angiopoietin receptor Tie1/2, HGF, and PECAM-1) in a cohort of cirrhotic patients for disease staging and therapy monitoring.

The analysis was conducted on a total of 29 cirrhotic patients, 18 women and 11 men, aged between 26 and 78 years; 22 out of 29 patients were diagnosed with metabolic cirrhosis and 7 with viral cirrhosis. The following comorbidities were observed: hypertension (*n* = 8), esophageal varices (*n* = 16), diabetes (metabolic, *n* = 17; insipidus, *n* = 1), portal hypertension (*n* = 7), the presence of ascites (*n* = 6), and encephalopathy (*n* = 7). The demographic and clinical parameters of patients are summarized in Table 1.

The possible use of angiogenesis markers for staging purposes is investigated in Figure 1, in which we study the correlation between these biomarkers and two widely used clinical scores, namely the MELD and FI scores.

Specifically, Figure 1a shows a correlation matrix of Spearman’s coefficients (ρ) computed between the investigated angiogenesis markers and the two mentioned clinical scores. The ρ values for a panel of conventional biochemical parameters are also included in the analysis, as will be discussed subsequently. Notably, the first seven rows of the matrix show the absence of significant correlation among any pair of angiogenesis markers, suggesting that different parameters are likely to provide independent information. Strong significant correlations are observed between angiopoietin-1/2 levels and the two clinical scores (lines 8–9, Figure 1a). No other significant correlations with MELD and FI are observed. For the sake of completeness, data leading to these significant correlations are visualized using scatter plots in Figure 1b–d. A linear regression model is fitted to the data and the best regression line is plotted on each graph together with 95% confidence intervals (gray area) and prediction bands (dashed lines). Additionally, the equation of the regression line with the corresponding R^2^ value is superimposed on each plot. Figure 1b,c shows the trend of Ang-1 as a function of MELD (ρ = −0.73, *p* = 2E−5, Figure 1a) and FI (ρ = −0.52, *p* = 7E−3, Figure 1a). The negative correlation of Ang-1 with the two clinical scores suggests that high levels of this marker can be protective against disease worsening. This hypothesis is in close agreement with the results of Pestana and coworkers [40]. Figure 1c shows the data of Ang-2 as a function of MELD (ρ = +0.53, *p* = 0.003, Figure 1a). An increase of Ang-2 levels is observed as the stage of the liver disease progresses, in close agreement with Hernández-Bartolomé and coworkers [41]. To get a better understanding of the trends shown in Figure 1b–d, it is worth recalling that the MELD and FI scores depend on several biochemical markers, namely creatinine, bilirubin, INR, albumin, and PLT. In this regard, further analysis of Figure 1a shows the presence of several significant correlations between Ang-1 levels and INR (ρ = −0.472, *p* = 0.012) and creatinine (ρ = −0.418, *p* = 0.034), and between Ang-2 and albumin (ρ = −0.526, *p* = 0.003) and DBIL (ρ = 0.728, *p* = 0.003).

Taken together, the results in Figure 1 confirm the potential applicability of Ang-1 and Ang-2 levels for the development of novel quantitative tools for cirrhosis staging.

A total of 7 out of 29 cirrhotic patients were diagnosed with viral hepatitis (HCV and/or HBV, Table 1). These patients were treated with direct-acting antiviral agents or HBV antiviral drugs, according to clinical guidelines (see Section 2). Patients underwent a second blood sampling at the end of the therapy, 8–24 weeks after SVR (see Section 2). The levels of angiogenesis and biochemical parameters were evaluated and compared with those measured before the treatment. In Figure 2, we showed the corresponding comparative analysis. Data are visualized using box plots. For each patient, points at different times are connected by a continuous line. The result of a Wilcoxon test for paired samples is superimposed on each graph. Despite the small sample size, a significant reduction after treatment was found for the Ang-2 and, consequently, for the Ang-2/Ang-1 ratio. No significant differences were found in the remaining angiogenesis markers. Taken together, the results of Figure 2 indicate the potential effectiveness of Ang-2 for therapy monitoring purposes.

### 3.2. Analysis of Angiogenic Biomarkers in Patients with Hepatocellular Carcinoma and Correlation with Clinical Parameters

Using a similar approach to that in Section 3.1, here, we aim at evaluating the clinical and prognostic significance of the investigated angiogenesis markers in a cohort of HCC patients. Analyses were conducted on a total of 33 subjects diagnosed with HCC (24 men and 9 women), aged between 49 and 85 years. In this cohort, 24 patients were affected by cirrhosis, 4 by alcoholic steatosis, and the remaining 5 showed no further liver disease. Twenty-seven patients had at least one other comorbidity. Specifically, 17 subjects showed signs of portal hypertension and 4 of deep vein thrombosis. Patients were subjected to different treatments, according to established clinical guidelines (see Materials and Methods). Specifically, six patients underwent hepatic resection, three underwent a transplant, four underwent pharmacological treatment with sorafenib, nine underwent local-regional treatment, and five received the best supportive care. At the end of the study, there were five deaths. Baseline parameters of the recruited subjects, including *exitus*, are summarized in Table 1.

Firstly—as was done for the group of cirrhotic patients without HCC—we evaluated the presence of significant correlations between the angiogenesis markers, clinical scores, and conventional biochemical parameters. Figure 3a shows a correlation matrix of the Spearman correlation coefficients, ρ, for each pair of variables. At variance with Figure 1a, in this case, we observe significant and positive correlations between different angiogenesis markers. On the contrary, Ang-1 shows a significant negative correlation with FI (ρ = −0.92, *p* = 0.001), in agreement with the results obtained for the group of patients without HCC (Figure 1c). PECAM-1 shows a positive correlation with the MELD score (ρ = 0.44, *p* = 0.01), which is likely related to its correlation with INR (ρ = 0.51, *p* = 0.002). Figure 3a also shows that, among the investigated markers, HGF (ρ = −0.38, *p* = 0.03) and Tie2 (ρ = −0.37, *p* = 0.03) display a mild negative correlation with age, a finding that will be further investigated in the following section.

Data of PECAM-1 as a function of MELD and Ang-1 as a function of FI are visualized in Figure 3b,c, respectively, using scatter plots. A linear regression model is fitted to the data and the best regression line together with confidence and prediction bands are superimposed on the plot.

The prognostic value of the selected markers of angiogenesis was evaluated considering the variable exitus in Table 1. Accordingly, patients were divided into deceased and not deceased during the study, and the angiogenesis markers’ levels were compared (Figure 4). Despite the small sample size, a statistically significant decrease of Ang-1 levels is observed in the deceased patients compared to not-deceased ones (Figure 4a). The performance of Ang-1 as a binary classifier is further evaluated through ROC curve analysis (Figure 4b). A significant area under the curve is obtained: AUC = 0.76 (95% CI: 0.57–0.96). Taken together, Figure 4a,b points out the possible effectiveness of Ang-1 as a prognostic index. To further test this hypothesis, in Figure 4c, we show the results of a logistic regression analysis of the two states as a function of Ang-1 levels. For this purpose, deceased/not-deceased subjects are indicated with 1/0. The continuous black line represents the probability of death as a function of Ang-1 computed from the regression coefficient: the lower the Ang-1, the higher the death probability. In this regard, it is worth stressing that the Ang-1 coefficient obtained from the logistic regression is associated with a *p*-value = 0.07. Therefore, data in Figure 4c need to be considered simply suggestive of the result, rather than statistically significant. Despite the lack of the 0.05 significance level, we decided to report this result as it agrees with and supports the results of Pestana et al., which showed how higher levels of Ang-1 are associated with longer overall survival [40].

The recruited HCC subjects underwent different anticancer treatments, according to well-established guidelines (see Section 2). After treatment, a second blood sample was collected from each patient and, both, angiogenesis and biochemical markers were evaluated and compared with values before treatments. Despite the large heterogeneity, a significant increase of VEGF was detected after treatment, a result that deserves a more in-depth study (Figure 5). Notably, a significant reduction after treatment is observed in AFP, which is one of the most widely used biochemical markers for HCC diagnosis and staging [38].

### 3.3. Clinical Significance of Angiogenic Biomarkers in the Progression of Chronic Liver Disease: A Comparison between Cirrhotic Patients and Cirrhotic Patients with Hepatocellular Carcinoma

HCC occurs, in most cases, in patients with liver cirrhosis rather than other conditions. The worsening of cirrhosis toward HCC is often related to increased pathological angiogenesis, which plays a key role in tumor growth. To assess the clinical significance of the angiogenic markers in liver carcinogenesis progression, in this section, we compare the expression of selected angiogenic markers in a cohort of 29 cirrhotic patients without HCC and 23 cirrhotic patients with HCC. For this purpose, HCC patients with underlying liver disease other than cirrhosis were not included in the analysis. Angiogenic markers in the two groups are compared in Table 2. A statistically significant increase in HGF levels is observed in cirrhotic patients with HCC. No other statistically significant differences are observed in the remaining markers. As far as HGF levels are concerned, *a caveat* is necessary. Figure 1a and Figure 3a show a dependence of HGF on the age of the patients. However, no statistically significant differences in age are measured in the two groups (*p* = 0.22). Data for HGF levels are visualized using a box plot analysis in Figure 6. Information regarding the age of the subjects was visualized using a color scale on data points.

The result of a Wilcoxon test for HGF levels and age is superimposed on the plot. Taken together, these results suggest that HGF levels can be investigated as a possible indicator of the progression of the disease from cirrhosis to HCC. This hypothesis could be validated by a longitudinal cohort study extender. For the sake of completeness, the comparison between blood levels of conventional blood markers is reported in Table 3. Only significant differences are shown.

## 4. Discussion

Hepatocellular carcinoma is the most common cancer affecting the liver, and its incidence almost entirely reflects its mortality. The high prevalence results from the high frequency in populations of the development of chronic liver damage, following hepatitis and/or cirrhosis. The initial lack of symptoms does not allow an early diagnosis and, therefore, a timely intervention to fight the neoplasm. In fact, HCC is a potentially curable form of cancer, but unfortunately, most patients present the disease at an advanced stage.

When a case of HCC is diagnosed at a relatively early stage, in which liver function is preserved, the most suitable approach and that which offers a higher rate of postoperative survival is surgical resection [42]. Despite the continuous progress of surgical techniques, of early diagnosis, the morbidity rate of patients undergoing liver resection remains very high. Therefore, compared to other types of solid tumors, the long-term prognosis (5 years after surgery) remains unsatisfactory, due to the high incidence of intrahepatic relapses [43].

In this scenario, scientists need to provide major benefits for the treatment of HCC. Targeting the hallmarks of cancer is usually one of the approaches to anchor this problem. For HCC, hallmarks include maintenance of proliferative signaling, avoidance of growth suppressors, escaping immune destruction, replicative immortality, promotion of inflammation, activation of invasions and metastases, inducing angiogenesis, mediating the instability and mutation of the genome, resisting cell death, and deregulating cellular energy [44]. This means that more hallmarks, more pathways, and cytokines are involved. It is, therefore, necessary to search for HCC markers that allow the identification and control of those who are at greater risk and that can also stratify patients according to the risk of cancer recurrence.

The progression of liver disease is accompanied by pathological angiogenesis, which is a prerequisite that favors the development of HCC. In HCC, hypoxia increases the expression of VEGF [45].^.^

As we showed in the results, VEGF levels significantly correlate with the levels of Ang-2 and HGF (Figure 3). In addition, following the treatment of HCC, the levels of VEGF were increased compared to the values reported at the time of diagnosis, as well as the levels of lymphocytes. This can be partly explained by the rebound effect of VEGF, induced by hypoxia following locoregional treatments, often associated with treatment failure and low survival rates in patients [46]. Interestingly, a reduction in AFP levels—a sensitive HCC biomarker—is detected after treatment.

Elevated Ang-2 levels have been reported in patients with HCC and cirrhosis. Increased Ang-2 levels have also been associated with advanced pathological features and worsening overall survival [47].

Patients with HCC enrolled for this study do not show circulating levels of Ang-1, Ang-2, and Tie1 and Tie2 receptors that are significantly different from those in patients with cirrhosis without HCC, but these molecules correlate with VEGF, CA 19-9, and platelets count in the case of Ang-1 and with PECAM-1 and HGF in the case of Ang-2. These data confirm the close relationship and interplay of these molecules in pathological angiogenesis and in the development of hepatocellular carcinoma.

Interestingly, we found a correlation between the levels of the Tie2 receptor and the levels of ALT, an index of cytonecrosis and surrogate marker of liver damage. The Tie2 receptor plays a key role during the late phase of angiogenesis and is responsible for the maturation of newly established vascular structures. This correlation highlights the parallel trend of hepatocellular damage and tumor angiogenesis in the progression of the disease.

Notably, patients with HCC show higher HGF concentrations than patients with cirrhosis (Figure 5, Table 3). These data confirm what has already been reported in the literature, namely that the expression of HGF and its receptor supports the existence of both autocrine and paracrine mechanisms of HGF action in HCC compared to the only paracrine mechanism in the liver without neoplasia, suggesting that it also plays a role in tumor development and/or progression [22,23,48]. We also found a strong correlation between HGF levels, Ang-2, and VEGF levels, further supporting the fundamental value of these markers in tumor angiogenesis.

The PECAM-1 has been found to correlate positively with MELD, and its identification can help not only in assessing the degree of tumor angiogenesis, which can indicate a rapidly growing tumor, but also in estimating the severity of the disease and the probable survival of patients waiting for a liver transplant. It has been shown that PECAM-1 promotes the formation of metastases by inducing the epithelium–mesenchymal transition in HCC by increasing the regulation of β1 integrin through the FAK/Akt signaling pathway [49].

In a recent wide-scale clinical study studying both HCC and cirrhotic patients, Pestana and co-workers showed that high Ang-1 levels are associated with an increase in the overall survival of patients diagnosed with HCC, showing that plasma Ang-1 is a potential diagnostic and prognostic biomarker in HCC [40]. Despite the lower sample size, data shown in Figure 4a are in close agreement with these findings as they show a significant reduction of Ang-1 levels in deceased subjects during the study, in comparison with not-deceased ones. Consistently with [40], our data (Figure 4c) are also suggestive of an increase in the probability of death with decreasing Ang-1 levels and a negative correlation with FI (Figure 2). Additionally, the prognostic value of this marker is confirmed from the ROC analysis in Figure 3b that, despite the small number of deceased subjects, shows a significant AUC value (0.76, 95% CI 0.57–0.96).

Serum levels of angiogenic biomarkers were also evaluated in patients with cirrhosis to identify the presence of possible statistically significant differences, of potential diagnostic interest, or useful for identifying specific molecular mechanisms underlying the genesis, development, and worsening of the disease. In addition, for cirrhotic patients, Ang-1 has a negative correlation with the FI while Ang-2 is positively correlated with FI. These results confirm the potential of these markers for the staging of cirrhosis. In addition, Ang-1 correlates negatively with MELD, suggesting that high levels of this biomarker can be protective against the progression of the disease.

Considering the changes in markers before and after treatment (that consists, in the case of patients with viral cirrhosis, in the administration of DAAs or other antiviral drugs), a significant reduction in serum Ang-2 levels and, consequently in Ang-2/Ang-1 ratio, was found. The reduction of serum Ang-2 levels can be considered an indication of the success of the antiviral treatment and, therefore, potentially used for monitoring the disease during the therapy.

In conclusion, angiogenesis is a very complex process essential for tumor growth and the formation of metastases, which involves a very large number of factors. This work, although limited by the small sample size and a large study group heterogeneity, suggests that serum angiogenesis markers—Ang-1 and Ang-2—may play a key role in the staging and monitoring of liver disease. Furthermore, the comparison between cirrhotic patients with and without HCC suggests that HGF levels are potentially useful for monitoring the insurgence of HCC after a cirrhosis diagnosis. High Ang-1 levels in HCC patients appear to have a protective role as well as prognostic significance. Unfortunately, the assessment of these angiogenic markers in common clinical practices is not suitable for each hospital up to now, as it is expensive and requires specialized personnel and equipment. Despite this limit, further investigations on the clinical significance of angiopoietins levels and other angiogenic, also as predictors of response or resistance to therapies, would be very interesting, especially in the era of immunotherapy and HCC-targeted anti-angiogenic drugs.

## 5. Conclusions

Pathological angiogenesis in liver disease represents a prerequisite that favors the development of HCC that we here studied through the assessments of serum angiogenic markers. We suggest that Ang-1/2 can helpfully contribute to follow liver disease therapy response, mainly high Ang-1 levels in HCC patients appear to have a protective role as well as prognostic significance. High HGF levels are potentially useful for monitoring the insurgence of HCC after a cirrhosis diagnosis.

## Figures and Tables

**Figure 1 cancers-14-00011-f001:**
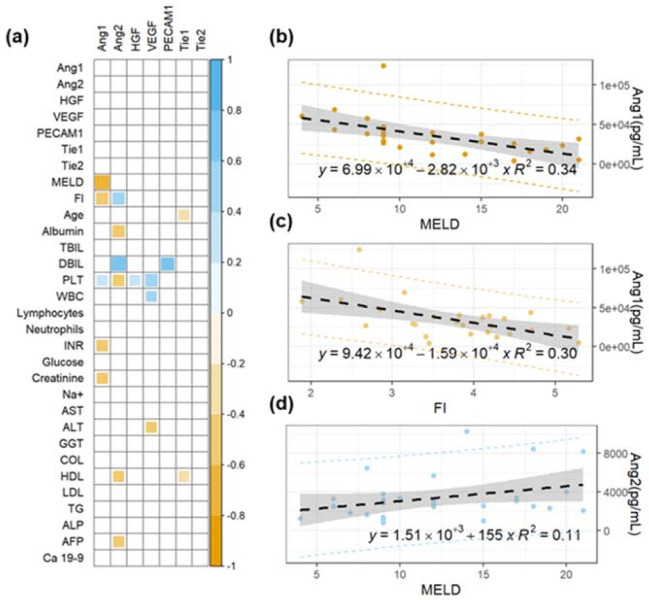
(**a**) Correlation matrix of Spearman’s coefficients, ρ, between the investigated angiogenesis markers and a panel of selected plasma biochemical parameters and clinical scores for cirrhosis staging (MELD and FI). A double color scale is used to assess the direction of the correlations, with negative values displayed in orange and positive ones in light blue. The strength of the significant correlation is directly proportional to the pixel size and the pixel intensity. Non-significant correlations are indicated as empty white pixels. Linear regression analysis of selected angiogenesis markers (angiopoietin-1 and angiopoitin-2) as a function of widely used clinical scores for cirrhosis staging (**b**–**d**).

**Figure 2 cancers-14-00011-f002:**
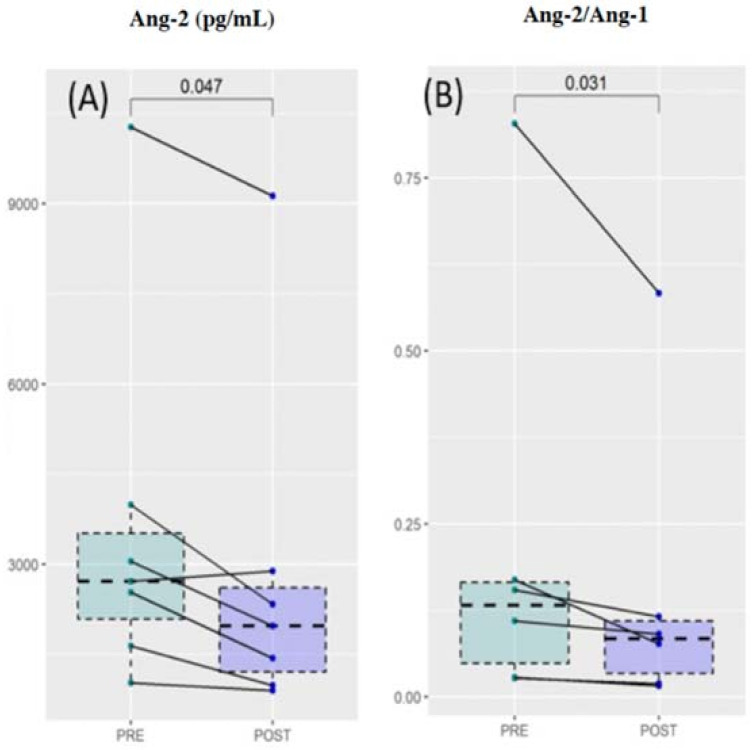
Comparative analysis of Ang-2 levels (**A**) and the Ang-2/Ang-1 ratio (**B**) before and after the treatment of patients diagnosed with viral hepatitis with the required antiviral treatment.

**Figure 3 cancers-14-00011-f003:**
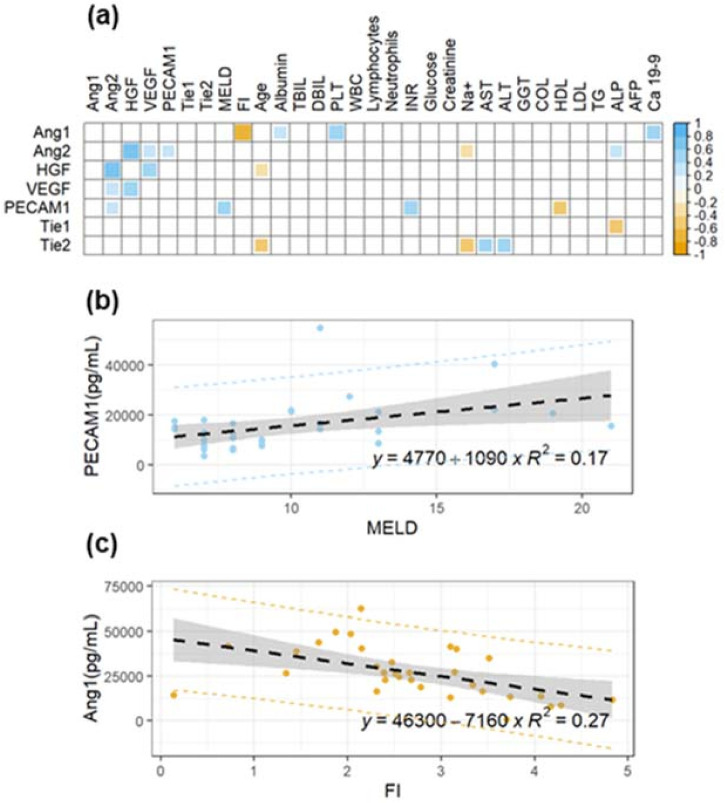
(**a**) Correlation matrix of Spearman’s coefficients, ρ, between the investigated angiogenesis markers and a panel of selected plasma biochemical parameters and clinical scores in a cohort of HCC patients. A double color scale is used to assess the direction of the correlations, with negative values displayed in orange and positive ones in light blue. The strength of the significant correlation is directly proportional to the pixel size and the pixel intensity. Non-significant correlations are indicated as empty white pixels. (**b**) Linear regression analysis of PECAM-1 as a function of MELD for the same subjects. (**c**) Linear regression analysis of Ang-1 as a function of MELD for the same subjects.

**Figure 4 cancers-14-00011-f004:**
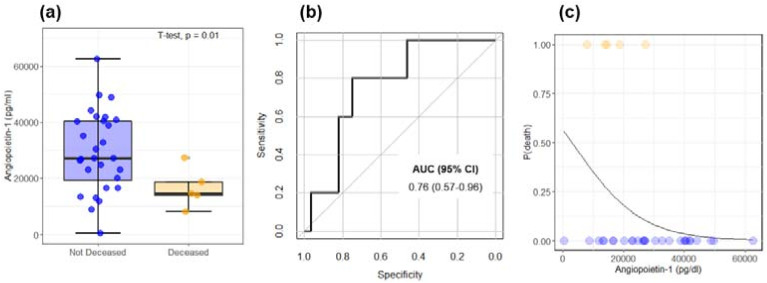
(**a**) Comparison between Ang-1 levels in deceased and not-deceased HCC patients. (**b**) Receiving operator characteristic curve computed from the data in Figure 3a for the evaluation of Ang-1 as a binary classifier. (**c**) The probability of death as a function of Ang-1 is computed from the coefficient of a logistic regression analysis.

**Figure 5 cancers-14-00011-f005:**
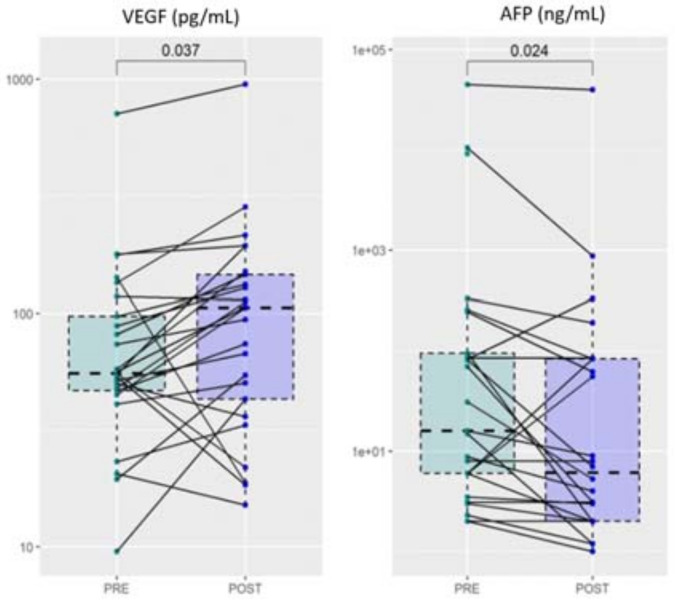
Comparative analysis of VEGF and AFP levels before and after HCC treatment.

**Figure 6 cancers-14-00011-f006:**
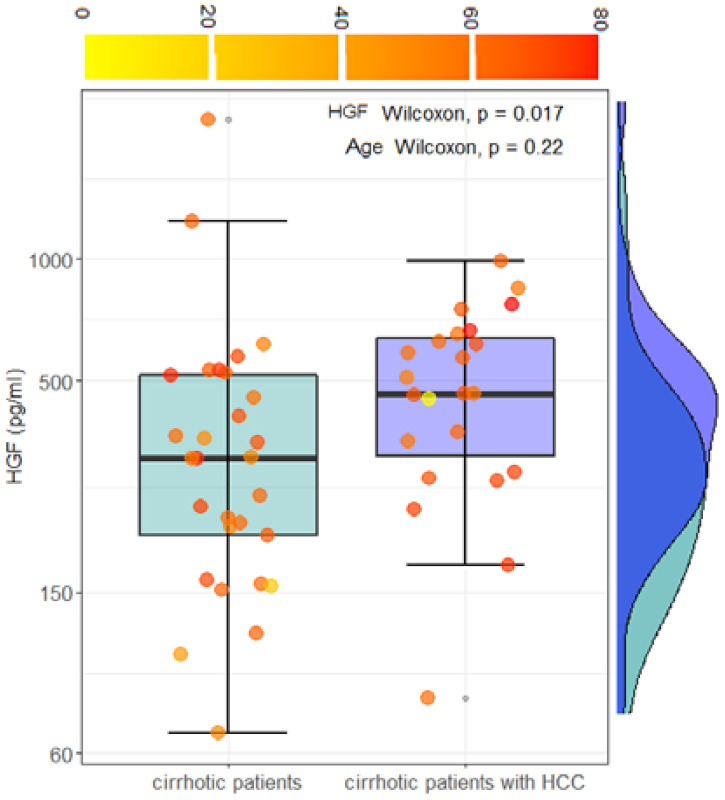
Box plot analysis and marginal distributions of HGF levels in cirrhotic patients and cirrhotic patients with HCC, before the treatments. Patients’ age is visualized using a continuous color scale.

**Table 1 cancers-14-00011-t001:** Clinical and demographic parameters of the study population.

Variable	Cirrhosis	HCC
**N^o^. of patients**	29	33
**Age** mean/median (IQR)	62/63(11)	67/67(15)
**Gender** (M/F)	11/18	24/9
**Underlying liver disease**		
Cirrhosis	29	24
Steatosis (NASH)	–	4
Healthy liver	–	5
**Etiology**		
HCV	4	11
HBV	2	5
HCV and HBV	1	1
Potus	2	7
Metabolic	7	8
Cryptogenic	13	6
**Portal hypertension**	7	17
**Esophageal varices**	16	9
**Nodule size (cm)**	–	
<2		8
2–5		14
>5		11
**Vascular thrombosis**	–	4
**Ascites**	6	7
**Encephalopathy**	6	2
**Diabetes**	18	10
**BCLC**	–	
0		4
A		14
B		4
C		3
D		1
**Child–Pugh**		
A	14	20
B	11	7
C	4	–
**MELD**		
0–9	13	20
10–19	13	12
20–29	3	1
**Treatment**		
HCV treatment (DAAs)	5	12
HBV treatment	2	5
Hepatic resection	-	6
Transplant	-	3
Sorafenib	-	4
Local–regional	-	9
Best supportive care	-	5
**Exitus** (death)	-	5

**Abbreviations:** HCC: hepatocellular carcinoma; HCV: hepatitis C virus; HBV: hepatitis B virus; BCLC: Barcelona Clinic Liver Cancer stage; MELD: Model for End-Stage Liver Disease score; DAAs: direct-acting antiviral agents; NASH: non-alcoholic steatohepatitis.

**Table 2 cancers-14-00011-t002:** Comparison of selected angiogenic markers between cirrhotic patients and cirrhotic patients with HCC.

	Cirrhotic Patients without HCC					Cirrhotic Patients with HCC					Wilcoxon	
**Angiogenic Markers**	** *n* **	**Mean**	**SD**	**Median**	**IQR**	** *n* **	**Mean**	**SD**	**Median**	**IQR**	**w**	** *p* **
**Ang-1** (pg/mL)	26	34,435	24,517	28,900	20,541	23	23,005	10,517	22,900	12,965	391	0.066
**Ang-2** (pg/mL)	29	3401.4	2312.83	2869.45	1322.51	23	3377.96	2113.74	2906.95	1680.	311	0.688
Ang2/Ang1	22	0.25	0.36	0.13	0.09	23	0.19	0.14	0.14	0.22	245	0.866
**HGF** (pg/mL)	29	411.45	414.51	320.93	309.30	23	500.32	224.58	464.56	310.90	203	0.017
**VEGF** (pg/mL)	29	91.91	73.99	69.75	72.22	23	92.63	66.63	56.36	91.04	312	0.701
**PECAM1** (pg/mL)	29	15,237	5415.34	14,483.60	9571.68	23	16,260.82	11,400.07	13,429.44	7652.7	369	0.522
**TIE1** (pg/mL)	29	9.82	3.79	9.11	5.17	23	9.57	7.10	7.52	7.75	412	0.152
TIE2 (pg/mL)	29	22.40	8.36	22.51	10.86	23	20.91	7.48	21.25	10.62	357	0.674

**Table 3 cancers-14-00011-t003:** Comparison of plasma biochemical markers between cirrhotic patients and cirrhotic patients with HCC (only statistically significant differences are shown).

	Cirrhotic Patients without HCC					Cirrhotic Patients with HCC					Wilcoxon	
**Plasma Markers**	** *n* **	**Mean**	**SD**	**Median**	**IQR**	** *n* **	**Mean**	**SD**	**Median**	**IQR**	**w**	***p*-Value**
**Albumin**	29	30.3448	6.48777	29	9	23	37.2174	6.68755	39	11	154.5	0.001
**COL**	29	134.241	39.5769	133	30	23	155.609	46.2884	144	44	216	0.031
**LDL**	29	78.9517	31.7226	76	32.4	23	95.087	31.8661	92	19	184.5	0.0062
**AFP**	29	3.76931	3.00572	3	4	23	667.313	2273.62	8.8	80	126	0.0001

## Data Availability

The data presented in this study are available upon reasonable request to the corresponding author.

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
