# Peer review of "A Comparative Study of Serum Angiogenic Biomarkers in Cirrhosis and Hepatocellular Carcinoma"

_cancers, 2021, doi:10.3390/cancers14010011_

Round 1
Reviewer 1 Report
The authors have incorporated the comments raised by the reviewer and thereby significantly improved the quality of the manuscript. The manuscript can now be accepted for publication. Congratulations!
Reviewer 2 Report
I would like to congratulate the authors for the major effort to improve the overall quality of the manuscript and adressing our issues.
This manuscript is a resubmission of an earlier submission. The following is a list of the peer review reports and author responses from that submission.
Round 1
Reviewer 1 Report
In this really interesting paper entitled “Biomarkers of angiogenesis in hepatocellular carcinoma: a novel sunshine road” Procino K et al. Provided an in depth molecular analysis of angiogenetic markers in both cirrhotic and HCC populations
I have a fee comments to potentially improve the paper
1) the study population is relatively small, and the Authors should stress this limitation in the discussion session
2) other limitations should also be discussed, such as the low clinical applicability of these markers in common clinical practice
3) the discussion on covid-19 is not supported by any of the results: despite covid-19 is a hot (and somewhat over-used) topic in current literature, I believe that such discussion section could be erased or shortened
4) consider citing this paper focusing on results of liver resection and ablation for HCC (10.1016/j.ejso.2019.04.023)
Best regards
Author Response
Dear Reviewer,
I attached the file with the answer to your suggesstions

Reviewer 2 Report
First of all, I want to congratulate the authors on the manuscript entitled “Biomarkers of Angiogenesis in Hepatocellular Carcinoma: A Novel Sunshine Road”. The group has successfully included 82 patients and measured serum concentrations of angiogenesis markers. I addition, they also measured serum concentrations of routine blood parameters. All angiogenesis parameters were correlated with blood parameters and demographics data and all measured parameters were correlated with the presence or absence of cirrhosis and/ or HCC and HCC treatment.
There are some minor and major comments that need to be addressed:
Title: The title sound like a title of a review on angiogenesis biomarkers in HCC. It would be better if it would reflect the actual study that has been conducted.
Abstract: Please indicate ALL study aims. You only state: “The aim of this study is to evaluate the clinical utility of circulating angi- 26 ogenic markers VEGF, Ang-1, Ang-2, the Angiopoietin receptor (Tie1/2), HGF and PECAM-1 to 27 screen early onset patients and to follow the evolution of HCC.“ In the abstract you don’t mention the findings of blood markers (except CA19-9). Either you need to report all aims and results, or you focus on one and then only report one aim.
Page 3 Patients: Can you please be more specific at the timepoints of the blood sampling?
When is the first visit? After diagnosis of cirrhosis or HCC?
We learned that patients with HCV had their second blood sampling after treatment – how long after treatment end (SVR? I guess?) was this sample taken (days, weeks, months?) This study represents a quite heterogenous group of patients (which is fine), yet, to make this study reproducible in any way, much clearer sampling timepoints need to be given. The manuscript would probably benefit from a schematic timeline for sampling in different etiologies (if not in the main manuscript, then as supplement) to clarify when biomarkers and routine blood samples were collected. This would also help later to interpret the results.
Also, you mention in the second paragraph patient with antiviral treatment and then in the fourth paragraph patients with DAAs. This should be mentioned in one paragraph to make it easier for the reader.
Page 4: Table 1: HCV e HBV – instead of e maybe and or &
Criptogenic = cryptogenic; please correct this spelling mistake
Nodule size (please indicate the unit e.g. cm)
Please add a footnote to all tables that explains all abbreviations used.
Portal hypertension: how was this defined? How did you measure this and what was your cut-off for saying that they had portal hypertension? Definition is missing in the method section.
Esophageal varices: how did you quantify them? When did you say that patients hat varices? Definition is missing in the method section.
Potus: what does potus mean?
Metabolic: does this refer to NASH/ASH?
Nodule size: did they all only have one nodule? I am missing the number of nodules
Etc. for Vascular thrombosis, ascites, diabetes (increased HbA1c, or oral treatment or insulin dependent?)
Please include a section in the methods with definitions of all variables that you assessed that need to be defined. There are different ways to assess some of these features and it is interesting how you defined those in your cohort. This will help to increase reproducibility and comparability for others.
Please include the info on different treatments for patients and please include the number of patients that died during the study (would be nice to have it all in one place and not just down in the text).
Demographical data (e.g. Table 1) as well as lab parameters (Table 2 and 3) are results and therefore should not be part of the method section. Please rearrange the manuscript accordingly. Also, you don’t mention in the aim nor the methods that you are assessing serum concentration of blood parameters (only the heading of Table 2 tells the reader that you are evaluating them). Please include this in the method section. Where they taken at the same time as the angiogenesis marker? Were the samples analyzed during clinical routine? Please specify.
Fibrosis index: please don’t use abbreviations unless the abbreviation has been used before. This as well as MELD could be defined in the method section, then it doesn’t interrupt the flow of the statistical section.
Figure 1A: There is no reference to this Figure in the text? Please include that and explain to a non-statistician what we see here.
Page 8 239-244 and 253-257: I am not sure if the analysis of laboratory markers is within the scope of this paragraph. The manuscript is shifting from an analysis of angiogenetic biomarkers in cirrhotic patients and correlation with clinical parameters (heading of this section) into a “what laboratory markers are significantly different between subsets of my group”. I don’t think that the number of patients and multiple subgroup analysis bring reliable results on potential biomarkers and thus would advise the authors to focus on their primary aim which is to evaluate biomarkers for angiogenesis in this setting.
Are there data on expression of biomarkers at different ages? The age gap in this cohort is very large and this might constitute a critical bias.
Figure 3A: Please explain this figure and the main findings to the reader. It is mentioned, but not explained.
Page 11 lines 300-303: Please indicate the n for each of the etiology subgroups.
Page 11 lines 311-318: I don’t understand the difference between the grouping here and the grouping in the paragraph on top? What three different groups stratified according to underlying liver disease did you create? Isn’t that the same as in the paragraph before just less groups? There is again the mix of angiogenesis biomarkers and more routine blood parameters. At least for me this is very confusing and really takes away a lot of readability. Maybe you are able to make this a separate subsection. I understand the correlation analysis of angiogenetic biomarkers to blood markers, but then you added a section where you perform different subgroup analysis for angiogenesis and blood markers. Firstly, to make this study more specific to the proposed aim, I think it would be enough to focus on your primary research question (see the aim you stated in the Abstract or end of your introduction). Otherwise, I suggest that you at least make this a separate section to clearly divide these sections. This will improve readability and help to find the sections of interest.
Figure 4. How did you define pre and post in patients with BSC? Did you take treatment response into account? There is quite some heterogenicity of treatments.
Page 12/13 lines 348-352: This is a very interesting finding that should not be just put in a final paragraph. The comparison of markers of patients with cirrhosis (without HCC) and with HCC as an indicator of malignant transformation is clinically highly relevant. Please add p-values (higher levels can just be “higher” or statistically significant?) and/or refer to a table where the values can be seen. What about the other markers? Did you only compare cirrhosis and HCC or all HCC patients (also those without a cirrhosis)? Is there a potential bias?
The discussion is, compared to the result section much clearer and is a good summary of the not so clear result section. Yet, it can be shortened and even though COVID-19 is present as ever, the last two paragraphs do not add value to the proposed manuscript. There are other aspect that should be highlighted more like the limitations. The limitations of this study ground on the small sample size, but should be highlighted more clearly. What about the huge difference in age between the enrolled patients? Are there data on different expression of markers at different ages? Did you adjust for this bias in your analysis? What about the different treatment modalities for HCC. I am sure that there is quite a distinct difference between a patient that receives a liver transplant and one on best supportive care. Did you account for different treatment modalities in your analysis? These are only two examples of major limitations of this work that need definitely better highlighting and discussion.
Page 2 Lines 54-56: Layout; please insert and between environment, angiogenesis.
Page 3 line 107: I kind of see the message of this sentence, but its not really well written. Please rephrase: “Stemming from this background, it could be possible to keep subjects at risk of HCC under control. “
Page 3 line 108: The underlined part of the sentence needs rephrasing: However, it is not always possible to carry out an accurate screening of the 108 population at risk of HCC, especially for the costs of the procedure would have.
Author Response
Dear Reviewer,
I attached the file with the answers to your suggestions

Reviewer 3 Report
Thank you for submitting your work. Pocino et al try to explore the potential biomarkers implicated in angiogenesis in cirrhosis and hepatocarinogenesis , by comparing a group of patients with cirrhosis without HCC to a group of patients with HCC (with or without cirrhosis).
Although the idea and the desing are attractive, the article fails to provide data that could be used in clinical practice. The main flaw is, in my opinion, the lack of a clear primary endpoint, and a subsequent inappropriate methodology.
Major issues:
-What is the primary endpoint of the study?
-The two groups are not comparable: the HCC group comprises cirrhotic and noncirrhotic patients, the male/female ratios are very different between the two groups and the severity of liver disease is too different between groups.
-Table 1 should be provided with mean/median values and IQR.
Minor issues:
-Needs proofreading, for example: page 7 line 226, there is an error, figure 2 does not correspond to the text mentionned; page 12 line 319-320: I am not aware of class D in the Child-Pugh classification.
-Some comparisons are statistically absurd: page 12 line 345, bilirubin is used in the MELD calculation so those two variables are expected to be colinear.
-Discussion is too long and there is no point to mention Covid-19 in the context.
-
Author Response
Dear Reviewer,
I attached the file with the answer to your suggestions.
